# Rare Diseases: Needs and Impact for Patients and Families: A Cross-Sectional Study in the Valencian Region, Spain

**DOI:** 10.3390/ijerph191610366

**Published:** 2022-08-19

**Authors:** Cristina Gimenez-Lozano, Lucía Páramo-Rodríguez, Clara Cavero-Carbonell, Francisca Corpas-Burgos, Aurora López-Maside, Sandra Guardiola-Vilarroig, Oscar Zurriaga

**Affiliations:** 1Preventive Medicine and Public Health, Hospital Universitario Doctor Peset, 46017 Valencia, Spain; 2Rare Diseases Research Unit, Foundation for the Promotion of Health and Biomedical Research in the Valencian Region, 46020 Valencia, Spain; 3Economic, Demographic and Social Statistics Service, Valencian Institute of Statistics, Valencian Region, 46020 Valencia, Spain; 4Directorate General for Public Health and Addictions, Regional Ministry of Health, Valencian Region, 46020 Valencia, Spain; 5Department of Preventive Medicine and Public Health, Food Sciences, Toxicology and Forensic Medicine, University of Valencia, 46010 Valencia, Spain

**Keywords:** rare diseases, diagnostic delay, unmet needs, primary care, Spain

## Abstract

Families with rare diseases (RDs) have unmet needs that are often overlooked by health professionals. Describing these needs and the impact of the disease could improve their medical care. A total of 163 surveys were obtained from patients visiting primary care centres in the Valencian Region (Spain), during 2015–2017, with a confirmed or suspected diagnosis of RD. Of the 84.7% with a confirmed diagnosis, 50.4% had a diagnostic delay exceeding one year, and it was more prevalent among adults (62.2%). Families with paediatric patients were in a worse economic situation, with lower incomes and higher monthly disease-related expenses (€300 on average). These expenses were incurred by 66.5% of families and were mainly for medication (40.3%). Among them, 58.5% reported not being able to afford adjuvant therapies. The disease had an impact on 73.1% of families, especially on their routine and emotional state. Expenses, needs, and impacts were more frequent among families of patients with a history of hospitalisation or deterioration. Patients with delayed diagnosis had a higher consumption of drugs prior to diagnosis. People affected by RDs in the Valencian Region need therapies to improve their autonomy and emotional state. Health professionals should be aware of these needs.

## 1. Introduction

In Europe, rare diseases (RDs) are defined as those that affect fewer than 5 in 10,000 people. It is estimated that there are between 6000 and 7000 different RDs, and most of them are of genetic origin. Although individually rare, together they affect approximately 30 million people in the EU and 920,000 people in Spain [1]. However, prevalence data are not consistent, both because of the lack of agreement on the definition and because of the difficulties in detecting and recording these pathologies [2].

Improving knowledge of RDs and the social and health care of patients and their families has long been a public health challenge. Some of the problems faced by these families, such as difficulty in accessing medicine treatment, delays in diagnosis, and even poor training of healthcare workers to detect these diseases, were highlighted in the 1989 report of the US National Commission on Orphan Diseases [3]. Subsequently, both the study carried out by the European EURORDIS Alliance in 2003–2008 on 12,000 patients [4] and the ENSERio study conducted in Spain in 2016–2017 [1], as well as research undertaken in other countries [5], confirmed that these problems still exist; in particular, almost half of the patients continue to experience a delay in diagnosis and lack adequate treatment. In fact, one of the goals set by the International Rare Diseases Research Consortium (IRDiRC) for the period 2017–2027 is the confirmation of diagnosis in less than one year [6].

Diagnostic delay and other factors have an impact on the daily lives of patients and their families that needs to be assessed, especially as, in some cases, this impact may be greater than that of other more prevalent chronic diseases [7,8]. In 2017, EURORDIS conducted the first survey to assess this, showing, for example, that for 73% of participants, the disease represents a high economic cost, and almost half of them are unable to improve their autonomy because therapies, such as physical medicine and rehabilitation and psychological support, are not covered [9]. Assessing the impact is not easy, partly, because it may vary depending on clinical factors, such as time of diagnosis, severity, and type of disease, as well as depending on certain social determinants of health, such as gender, age, socioeconomic status, and type of health system. Any of these factors could generate inequities in access to services and, therefore, a greater impact on the lives of these families.

Furthermore, studies in the field of RDs are often carried out on specific pathology groups, which may prevent a comprehensive definition of the problems of this group: a problem also shared by those who still have a suspected diagnosis. Primary care professionals, although they generally see a small number of patients with RDs, may be confronted with a wide variety of conditions [10]. Moreover, due to the nature of their care, they often have to respond not only to clinical needs but also to social and other healthcare needs [11]. For all these reasons, in this study, we chose to begin with those patients and families seen in the primary care with a confirmed or suspected diagnosis of RD in order to understand their needs and better grasp the impact the disease has on their lives. 

## 2. Materials and Methods

A cross-sectional study was carried out based on a semi-structured questionnaire provided to families with RDs living in the Valencian Region (VR), Spain, with the aim of describing their socio-demographic characteristics, needs, and the perceived impact of the disease in different areas of life, such as social, occupational, and emotional.

The VR is one of the 17 Spanish autonomous communities, with 5,058,138 inhabitants in 2021, which represents approximately 10.7% of the population of Spain. The distribution by sex and age is similar to the national distribution, with women accounting for 50.7% and the 16–64 age group accounting for 64.6% [12].

The study is part of the project “Analysis, determinants and impact of diagnostic delay in rare diseases”. The aim of this original project was to calculate the diagnostic delay and to evaluate the possible impact on families. This project was conducted in two phases: a first quantitative phase, based on the questionnaire used for this study, and a second qualitative phase, in which focus groups were held with patients and professionals. The development of the qualitative part took place after the start of the COVID-19 pandemic, so its impact on patients and families could also be evaluated. This original project obtained the approval of the Ethic Committee of Clinical Research of the Directorate General for Public Health and Higher Center for Research in Public Health. All participants agreed to participate by signing the informed consent form for this study.

Sample selection: Patients with a diagnosis (confirmed or suspected) of RD, identified by primary care professionals, who participated in the VR Health Sentinel Network (HSN) during 2015–2017 were selected; the functioning of the HSN is detailed in the article by Vega et al. [13]. The definition of rare disease adopted by Europe was used. Patients without diagnostic confirmation were included when the main diagnostic suspicion was a rare disease and/or diagnostic tests for any of these diseases had been requested. From October 2019 to March 2020, the questionnaire and informed consent were mailed to families with a confirmed or suspected diagnosis of RD. Individuals belonging to various RD associations in the VR were also invited to participate by making the questionnaire available during the holding of conferences or through their websites. Subsequently, a validation process was carried out on the questionnaires received. For the analysis of the results, patients whose usual residence was not in the VR were excluded.

Questionnaire: A semi-structured, self-completed questionnaire (available in the Appendix A) was used, which asked 52 questions divided into three sections: the first with socio-demographic data of the patient, the second with questions on the economic and employment situation of the family unit, and the third with questions related to the diagnosis of the RD. The questions in the second section were adapted from a previous questionnaire developed by the working group of the Spanish multicentre project INMA (“Childhood and Environment”) [14]. That questionnaire calculated the AROPE index (used in the Horizon 2020 programme and the 2030 Agenda to measure the risk of poverty and social exclusion) based on the sub-indicators “income”, “job intensity”, and “material deprivation”. These variables were also included in the second section of our questionnaire. The third section asked about the diagnostic status, i.e., the presence of a confirmed or unconfirmed diagnosis and the time elapsed until diagnosis, as well as the different treatments received both before and after diagnosis. In order to specifically describe the treatment received, the questionnaire differentiated four groups: drug therapy (medicine); other treatments (including healthcare products—such as emollient creams and glasses—and therapies such as physiotherapy and rehabilitation); alternative therapies (such as acupuncture and other practices that are not part of conventional medicine); medicinal plants and plant-derived products. Finally, multiple-choice questions were included to assess the existence and distribution of disease-related expenses, resources not available in the healthcare system and not affordable for the household budget, and the emotional and social impact.

Description of data and statistical analysis: A descriptive analysis of the data obtained from the questionnaire was carried out, firstly by performing a univariate analysis, using measures of central tendency and dispersion for the quantitative variables and frequency distribution for the qualitative variables. One of the variables used to describe the economic situation of the families was material deprivation, defined as the absence of at least three of the nine resources considered desirable to achieve a good quality of life [15]. Subsequently, a bivariate analysis was performed to analyse the possible relationship of the variables, assessing the costs, needs, and impact of the RD with factors such as sex, age, diagnostic delay, and severity. To assess the relationship with age, the categorical variable “type of patient” was created, which defines a patient as paediatric if their age is between 0 and 14 years and adult if they are 15 years or older. The categorical variable “diagnostic delay” was defined as the interval, greater than one year, between the onset of symptoms and confirmation of diagnosis, thus using the same criteria as IRDiRC for its targets to be reached in 2017–2027. To define clinical severity, the categorical variables “hospitalisation” (which refers to the history of hospital admission as a consequence of the RD) and “aggravation” (whether there has been an aggravation of symptoms since the diagnosis) were used.

## 3. Results

A total of 691 cases were identified from the VR HSN and RD associations. After sending the questionnaires and going through the validation process, 177 cases were obtained, of which 14 were rejected because they related to people who were not residents of the VR or who had not completed the informed consent form.

### 3.1. Socio-Demographic Characteristics

Of the 163 questionnaires included in the study, 75 (46%) were completed by the patient him/herself, 61 (37.4%) by the mother, 19 (11.7%) by the father, and 8 (4.9%) by other relatives.

Of the patients, eight (4.9%) were of foreign origin. The mean age of the patients was 31.5 ± 23.2 years. Women represented 58.3% of the sample, with a mean age of 37.7 ± 22.6 years, and men represented 41.7%, with a mean age of 22.9 ± 21.3 years (*p* < 0.05).

By patient type, 63.2% were adults over 14 years of age, of which females accounted for 69.9%, while paediatric patients up to 14 years of age accounted for 36.8% of the sample, with the majority being male (61.7%) (see Figure 1).

### 3.2. Socio-Economic Situation of Families with Rare Diseases

The questionnaire measured the socio-economic situation of the families in terms of net monthly income, the existence of difficulties in making ends meet, the capacity to cope with possible expenses derived from the RD, and the risk of social exclusion due to material deprivation.

Of the 163 households included, 157 provided information on monthly income. More than half of these families (54.1%) did not exceed €2000, a proportion that rises to 65.5% of families with paediatric patients, compared to 47.5% of families with adult patients (*p* < 0.05). No statistically significant differences were observed between sex, the existence of a diagnostic delay, or a history of hospitalisation or aggravation.There were 138 families who responded to the question about having difficulties in making ends meet, with 33.3% reporting such problems. Statistically significant differences were detected depending on the type of patient (44.2% of families with paediatric patients compared to 26.7% of families with adult patients, *p* < 0.05) but not according to sex, the existence of a diagnostic delay, hospitalisation, or aggravation.There were 148 families who answered the questions related to their ability to pay for possible expenses arising from the RD. Among them, 62.2% of the 148 families could not afford, if needed, other treatments, private psychological therapy, or a caregiver, and up to 36.5% could not afford any of these three needs. The most frequent payment they could not afford was that of a caregiver (60% of the families), with statistically significant differences by type of patient (75% of families with paediatric patients and 51.1% of families with adult patients, *p* < 0.05) but not by sex, the presence of a delay, or a history of hospitalisation or aggravation.Material deprivation could be estimated in 158 families and was detected in a total of 17 households (10.8%): 16.7% of families with a history of hospitalisation compared to 4.3% in families without such a history (*p* < 0.05). No statistically significant differences were found by type of patient, sex, the existence of a diagnostic delay, or aggravation.

### 3.3. Diagnostic and Clinical Features

In terms of diagnosis, 138 patients (84.66%) had a confirmed diagnosis of RD, with no statistically significant differences by patient type or sex (*p* > 0.05).

The time from symptom onset to diagnosis could be calculated in 129 patients, with a median of 14 months (IQR 2–65), with five (IQR 0–18 months) in paediatrics and 23.5 (IQR 5.75–95 months) in adults (*p* < 0.05), with no differences by sex.

Among patients, 50.4% suffered a diagnostic delay according to the definition suggested by IRDiRC (more than one year from the onset of symptoms to the confirmation of diagnosis), with a higher proportion of adult patients than paediatric patients (62.2% and 29.8%, respectively, *p* < 0.05), with no differences by sex and with a mean age at diagnosis of 28.92 ± 21.39 years. Half of the patients with delayed diagnosis had been diagnosed after more than five years, and up to 25% were diagnosed after more than eleven years. Table 1 shows the data disaggregated by type of patient and sex.

Of the 138 patients with a confirmed diagnosis, 47 (34.06%) had a diagnosis of a condition from the ICD10-ES group of “Congenital malformations, deformations and chromosomal abnormalities Q00–Q99”, with this percentage being 62% of paediatric patients. Within this diagnostic group, 20 cases corresponded to the subgroup “Other congenital malformations Q80–Q89”.

There were 25 patients (18.12%) who had a diagnosis of a condition from the group “Musculoskeletal and connective tissue diseases M00–M99”. All of them adult patients, for whom this group of diseases was the most frequent, representing 28.41% of the diagnoses. All but one of the conditions belonged to the subgroup “Systemic connective tissue disorders M30–M36”.

There were 18 patients (13.04%) who were diagnosed with a condition from the group “Diseases of the nervous system G00–G99”, with half of the cases belonging to the subgroup “Diseases of the neuromuscular junction and muscle G70–G73” (see Figure 2).

The clinical progression was defined by the variables “history of hospitalisation” and/or “aggravation” (non-exclusive), to which 159 and 158 patients responded, respectively. Among patients, 84 (52.83%) had at least a history of hospitalisation, and 68 (43.03%) had at least a history of aggravation. In 48 patients (30.18%), both circumstances were present. No statistically significant differences were found by sex, type of patient, or presence of diagnostic delay.

### 3.4. Treatment

Regarding drug therapy for the RD at the time of the survey, of the 142 patients with information, 96 (67.6%) were receiving treatment with one or more drugs, with no statistically significant differences between those with a confirmed diagnosis and those with a suspected diagnosis (65.6% vs. 85.7%, *p* > 0.05). Treatment was chronic for more than half (85 patients, 59.9%).

Considering only those patients with a confirmed diagnosis, the use of any type of treatment was analysed both before and after diagnostic confirmation. In the latter case, the proportion of patients receiving drug therapy and/or other medical treatments increased compared to the stage prior to diagnostic confirmation (from 32.4% to 64.9% and from 33.1% to 51.1%, *p* < 0.05, respectively), but the proportion of patients using alternative therapies or herbal products did not change significantly (see Figure 3).

When comparing the use of these treatments, both before and after diagnostic confirmation, according to sex, patient type, diagnostic delay, and severity, the following results were obtained (see Table 2):Adult patients used drugs more frequently than paediatric patients, both before and after diagnosis.Patients with a diagnostic delay used drugs and other treatments in the stage prior to diagnostic confirmation more frequently than those without a delay, but after diagnostic confirmation, no statistically significant differences were found in any of the treatments evaluated compared to those without a delay.Patients with clinical aggravation consumed more drugs and other treatments both before and after diagnostic confirmation. After diagnosis, they consumed herbal products more frequently than non-aggravated patients.Patients with a history of hospitalisation used alternative therapies more frequently after diagnosis than those without prior hospitalisation.No statistically significant differences by sex were found.

### 3.5. Expenditures and Unmet Needs Arising from Rare Diseases

There were 161 families who provided information on expenses secondary to the disease. Among them, 107 families (66.5%) incurred at least one RD-related expense, and this proportion was higher for those with a history of hospitalisation (78.6% vs. 52.7%, *p* < 0.05) or aggravation (80.9% vs. 54.4%, *p* < 0.01), with no significant differences by patient type, gender, or delay. The median monthly expenditure was €255 (IQR 100–500), and was slightly higher for paediatric patients than for adults (€300 vs. €200, *p* < 0.05).

The most frequent expense was pharmacological (40.3% of families) (see Figure 4), with this being more frequent in women (50% compared to 26.9% of men, *p* < 0.05), in adult patients (40.3% compared to 27.1% of paediatric patients, *p* < 0.01), in patients with a history of a delayed diagnosis (53.8% compared to 33.9% of patients without a delayed diagnosis, *p* < 0.05), and in those families with a history of hospitalisation (51.8% vs. 28.4%, *p* < 0.05) and aggravation (52.2% vs. 31.1%).The rest of the expenses, with the exception of specialised care centres, were more frequent among families with aggravation, and expenses for psychological therapy and alternative medicine were also more common among families with a history of hospitalisation (see Table 3).

**Figure 4 ijerph-19-10366-f004:**
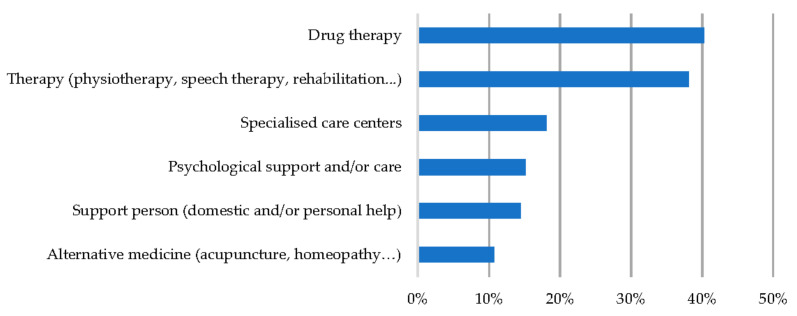
Percentage of households with disease-related expenses.

Ten possible needs generated by RDs, not covered by the Spanish public healthcare system and which the families could not afford, were assessed. Information was obtained from 159 families, of which 93 (58.5%) had at least one need not covered by the health system, which is a percentage that rose to 73.5% among patients with a history of hospitalisation and 82.1% among patients with aggravation, compared to 43.2% and 42.1% among patients without such a history (*p* < 0.01). No statistically significant differences were found by type of patient, sex, or presence of delay. Up to 22.2% of the families had more than three needs. The most frequently unmet need was access to non-clinical adjuvant therapies (42 families, 26.4%) (see Figure 5).

All needs, with the exception of health care and medicines, were more frequent among patients with clinical aggravation. Physiotherapy, adaptation of housing and workplace, non-clinical adjuvant therapies, and leisure were also more frequent needs among patients with a history of hospitalisation.Lack of medicines or healthcare products was more common among patients with diagnostic delays, where it was also the most frequent need (27.7%).No differences were found by type of patient or gender in any of the needs analysed (see Table 3).

### 3.6. Impact on the Emotional and Social Sphere

There were 155 families who responded to questions assessing eight possible consequences of RDs in the social, occupational, and emotional spheres. In 126 families (78.8%), there was at least one effect in some of the areas assessed, with this proportion being higher among families with a history of hospitalisation (91.7% vs. 63.5%) and aggravation (94.1% vs. 66.7%) (*p* < 0.01) but with no differences by sex, type of patient, or existence of delay. The most frequent impact was disruption of daily routine (59.4% of families) (see Figure 6). Families with some impact explained it, in most cases and at least in part, as being caused by the disease itself (73.1%), rather than by the delay in diagnosis (32.5%). However, the delay in diagnosis could explain the social and emotional impact, at least partially, for 49.2% of the families who have suffered a diagnostic delay, according to the IRDiRC definition, compared to 17.5% of those who received a diagnosis in less than a year (*p* < 0.01).

Among the effects evaluated, the need to travel to other health centres could be the one most likely to be explained by a delay in diagnosis (up to 21.9% of families say this is the case).

All the situations assessed were more frequent in families with a history of hospitalisation and aggravation. Mood disorders were also more frequent in women (65.2% vs. 48.5% of men, *p* < 0.05) and adult patients (65.3% vs. 45.8% of paediatric patients, *p* < 0.05). No statistically significant differences were found for the presence of diagnostic delay (see Table 3).

## 4. Discussion

This study has sought to portray the characteristics of families with RDs that a general practitioner may encounter in his or her practice, what their needs are, and how the disease impacts them by recording their perceptions in a semi-structured survey. Although paediatric and adult patients differ in several aspects, such as the type of disease and other factors, such as diagnostic delay, both groups were included in order to obtain a global vision of the problems faced by all those affected.

In general terms, the results obtained are in line with those presented in international studies, such as EURORDIS, and with other national studies, such as the ENSERio study in Spain, and with those of countries such as Australia and France, among others [1,4,5,16]. The findings confirm that RDs involve a financial outlay, which is mainly spent on drug therapy and other healthcare products, and other necessary resources are not being accessed because they are often not covered. Although the portfolio of common services of the National Health System includes financed access to primary, emergency, and specialized care (including mental health care and rehabilitation), as well as orthoprosthetic material and pharmaceutical services (the latter with different degrees of economic participation), public resources are limited and do not cover all the services requested by users. These are, in most families with members with rare diseases, services that could improve patients’ autonomy and promote their social inclusion, such as adjuvant therapies and psychological care, and whose coverage by public healthcare systems has not yet been effectively implemented [17], despite the fact that they are necessary for the comprehensive care of those affected, as reflected in the Spanish RD national strategy [18,19]. Both these shortcomings, as well as their impact on family and personal dynamics, were more frequent in the most severe patients. However, in contrast to the findings of previous literature, they were not generally more common in patients with delayed diagnosis [1,4,20]. Delayed diagnosis is, nevertheless, a problem that affected more than half of the participants and whose impact can be added to that caused by the disease itself.

The sample was similar to that of previous studies of the RD population, with a majority of adult patients and a high prevalence of congenital and musculoskeletal anomalies, and with people without a confirmed diagnosis accounting for more than 10% (15% in our study, which was somewhat higher than the 11% in the ENSERio study). Slightly more than half of the people required more than one year for diagnostic confirmation, with this percentage being higher among adults, which confirms that delay continues to be a problem. Although it varies from one disease to another, diagnostic delay tends to be greater in adults [1,21,22]. People over 64 years of age were under-represented (only 8%), considering that almost half (45.79%) of people with RDs in Spain are over 65 years of age; however, this is common in studies based on voluntary participation, and it should be addressed on future occasions to try to increase their participation and, thus, obtain results that are more representative of the population [23].

The data obtained suggest that the financial situation of the households surveyed was similar to that of the general Spanish population, both in terms of monthly income (54.1% of the families surveyed did not exceed 2000 euros, compared to 52.52% of Spanish families in 2019) and if we consider the risk of social exclusion with material deprivation (10.8% in our sample compared to 11.8% of Spanish families in 2019) [24]. However, to conclude that RDs are not affecting the economic status of families, it would be necessary to carry out a study that analyses the level of income or the material deprivation, according to variables such as the age or the type of household, in order to compare the results with greater reliability.

Even considering that monthly household income may be comparable to that of the general population, it should be borne in mind that the rare disease entailed a financial cost for 66.5% of the families surveyed, and this can drastically reduce their purchasing power since, according to some studies, the costs associated with the disease can amount to 20% of income [1]. The most frequent expense was the purchase of medicines. This seems logical if we take into account that more than 73% of adult patients with RDs were taking them at the time of the survey, compared to 57.91% of Spanish adults, according to the 2020 European Health Survey, and only 34.57% of those affected by RDs in the VR were fully covered [1,25]. As no differentiation has been made between prescription and non-prescription drugs, this could mean, in addition to a lack of coverage or partial coverage, self-medication by those affected. In addition to pharmaceutical expenditures, households often spend part of their income on physiotherapy and other therapies, such as rehabilitation or speech therapy, at a similar rate to that reported in the ENSERio study. According to the national health strategy on RDs, these therapies are usually offered by the public health system to patients with an acute illness and not so readily to those with chronic diseases, as would be the case for those affected by RDs [19]. According to our data, approximately 15% of families pay for psychological care. Given that, according to the European Health Survey 2020, 4.77% of Spaniards visited a psychologist in the last year [25], it is safe to assume that those affected by RDs are regular consumers of psychological therapy. However, despite being included in the portfolio of services, access times are too long, and professionals are scarce [17], meaning that patients must ultimately turn to the private sector. These additional, unreimbursed costs only generate further inequalities within a group that already has more difficulty accessing resources than those affected by other, more prevalent chronic illnesses.

Even considering the outlay made by households to cope with RDs, more than half do not have access to certain resources. Although expenditures are mainly for the purchase of medicines, the most frequently neglected therapies are non-clinical (58.5%). While it is true that there is insufficient scientific evidence for some of these complementary therapies, others, such as adapted sports or even music therapy, can promote rehabilitation, pain control, and social integration in those affected by various conditions [26,27,28]. It would, therefore, be of interest to lower the economic barriers and take them into account for comprehensive care. Physiotherapy, which, as already indicated, represented one of the most common expenses, has also been one of the most frequently unmet needs, as in other studies [29]. The same is true for psychological care: more than 22% of families claim to need it and yet cannot afford it, leaving these families more exposed to the deleterious consequences of emotional disorders, such as anxiety and depression, and with fewer tools, if possible, to integrate into society [7]. The percentage of families who do not have access to leisure activities is not negligible either (20.1%). According to the ENSERio study, up to 44.4% of Spanish families are dissatisfied with their leisure activities, which is a situation that could be improved by inclusion in the portfolio of services, activities, or programmes aimed at the infirm [30].

Almost 80% of the households surveyed reported some negative repercussion or impact on their lives as a result of the disease. Measuring impact is complex, as it can be approached from multiple perspectives and must take into account certain factors, such as access to adequate treatment or the lack thereof. However, as our work is part of a research project whose main objective was to measure delay in diagnosis rather than adequacy of treatment, it was decided to assess the impact of both the disease itself and the possible delay in diagnosis from a social and emotional perspective. More than half of the families reported that their daily routine was disrupted, and while the ENSERio study specified causes, such as the patient’s physical disability, preventing them from performing basic activities, our study was more non-specific in an attempt to include other causes. In addition, more than half of the respondents (58.1%) claimed to suffer from mood disturbances, a figure somewhat higher than those reported by RD association studies, which found that 46% required help to overcome these emotional disorders [1,9]. The presence of anxiety and/or depression is obviously associated with a lower quality of life [31] and may lead to increased isolation with a worsening of social relationships, already impaired in 50% of the families surveyed. Some studies have shown that social support (low for 31.9% of respondents) is of great importance in helping to manage family stress [32]. As a result, families with RDs enter a vicious circle from which it can be difficult to escape without adequate support.

The impact on work, both for the patient and their caregivers, has been widely described in other studies and was reported by 40% of respondents in our study. However, only 13.2% expressed workplace adaptation as a need not covered by the system. This apparent incongruence could be explained by the lack of knowledge that those personally affected have of their social rights, having taken sick leave and/or been dismissed, but not having been informed of alternatives such as job adaptation.

With regard to perceived discrimination, the prevalence in our study was much lower (23.8%) than that reported by RD associations such as FEDER, which put it at 76% of patients. This may be due to the fact that, in our case, the respondents did not identify all the areas in which discrimination may actually occur, such as education or employment, which are specified in other studies [1].

For the majority of participants surveyed, the disease itself, rather than the delay in diagnosis, has been the cause of the impact in the different domains assessed. In fact, in a study by Bryson et al. asking the open-ended question “What are the main challenges of living with a RD?” Symptoms and limitations were mentioned as the main obstacles [33]. Therefore, diagnostic delay could behave as yet another of the multiple challenges faced by people with RDs, with our results suggesting a notable role for journeys to other healthcare facilities (for more than 20% of respondents). This would confirm the “diagnostic odyssey” [34] suffered by people with RDs, who have to visit numerous specialists in centres often far from home, for periods of time unacceptably long for other more prevalent chronic diseases, until a definitive diagnosis is obtained [35].

In order to determine whether certain factors, such as gender, age, and a diagnostic delay of more than one year, could be acting as possible determinants of inequity, the survey results were analysed in terms of these factors. In addition, given the heterogeneity of the sample and the fact that, due to the sample size, it was not methodologically advisable to analyse the variables by pathology group, severity was taken into account and defined indirectly through a history of hospitalisation or aggravation.

In general, there were no major differences by sex, except for expenditure and current drug treatment, which were more frequent in women (although in the latter case, the differences were not significant), but this is often the case in the general adult population [25,36]. Mood disorders were also more frequent in women, but unlike in other studies, no greater impact on employment or higher perception of stigmatisation was found [33]. However, it is noteworthy that it was mothers who most frequently completed the survey of paediatric patients, supporting the fact that even today the role of primary caregiver is still predominantly female [37].In terms of age, the only significant differences found relate to the worse financial situation of families of paediatric patients (with lower monthly income and somewhat higher disease-related expenses) compared to the households of adult patients, perhaps because in the latter cases some family members, or the patient, receive a pension and a treatment allowance.As we had anticipated, patients with a delay in diagnosis of more than one year did not have worse outcomes, except for poorer access to drugs and higher pre-diagnosis consumption. This could be due to the fact that, in the absence of a diagnosis and specific treatment, symptoms are not controlled and thus the patient would initiate a relentless search for different treatments [1,38].Finally, it seems that the consequences of RDs have been more frequent in patients who have had a more torpid evolution, with previous hospitalisations and/or aggravation of the disease: all expenses and needs have been more frequent (except for the need for pharmaceutical products and health care, generally covered by the system for dependent patients or pensioners); they have had a higher consumption of medicines and even of alternative therapies after diagnosis; in short, they have experienced a greater impact. Although previous studies also seem to indicate that the use of alternative therapies may be more frequent in chronic patients with a more advanced stage of their disease, others have found no differences in the use of alternative therapies [39,40].

The analysis of the surveys has made it possible to highlight certain deficiencies in the public health system that could be filled with a different approach in the following regional and national health plans: On the one hand, a redistribution of economic and personal resources could improve the availability of psychologists in primary care centres. This is a claim with a long history and also shared by those affected by more prevalent diseases. On the other hand, it would be interesting to promote the clinical follow-up of those patients without a confirmed diagnosis in order to prescribe a treatment as targeted as possible to their symptoms and avoid self-medication.

One of the limitations of our study is the sample size, which prevented us from analysing the results by subgroups, i.e., by pathologies. However, given that the problems of RDs tend to be shared by the majority of those affected, and that the aim of this study was not to establish differences according to clinical symptoms but to other factors, as well as to describe the profile of patients who may attend primary care consultations, we consider that the overall analysis carried out has allowed us to gain a better understanding of the reality of those affected. Another important limitation has been the absence of information on income or benefits received due to incapacity. These questions could be included in subsequent studies in order to analyse another dimension of RDs and to be able to interpret the data on the socio-economic level of the families more solidly.

On the other hand, we consider that one of the strengths of the work has been the selection method used, starting with primary care patients, which provides an opportunity to participate for people who, for various reasons, are less involved in associations and who are, therefore, usually excluded from this type of study. Moreover, surveys aimed directly at the people affected help us to understand their real needs better than those aimed at health professionals, who are not always aware of the difficulties and resources that families require [41].

## 5. Conclusions

Beyond the specific and differentiating features of the different RDs, sufferers share a common situation, characterised by difficult access to early diagnosis and treatment, as well as poor adaptation of social and healthcare resources to their clinical, emotional, and occupational needs, with the impact this entails [1,9]. This is the case even in places with a universal public health system, such as Spain, and more specifically in the VR. In this region, those affected by RDs need equal access to therapies that could improve their autonomy, such as physiotherapy and rehabilitation, and to psychological therapies that could help them cope with the emotional burden of the disease, as well as earlier diagnosis that could reduce the, perhaps improper, consumption of medicines. The limited availability of these services from the public system increases inequity in this group, perhaps with a greater impact on the families of paediatric patients. Although it is the patients with the most severe progression of the illness who seem to have the greatest needs and experience the worst impact, all could benefit from a socio-health approach to their pathology.

The training of health professionals in the field of RDs, which is generally insufficient, should, therefore, include not only the clinical aspects, which can obviously be specific to each condition, but also these socio-health aspects, which would be very effective because they can be applied to all those affected. It is, therefore, a matter of achieving comprehensive training that will enable professionals to make the existing resources available to their patients and to help them while the healthcare systems adapt [8,11,42]. This would be especially important for primary care professionals, who are often the first point of contact, to whom the chronic follow-up of patients is sometimes delegated, and who could coordinate resources, provided they are aware of the needs of families with RDs [43,44]. Although the figure of the social assistant exists in the Spanish public system, the health professional can provide an integrating vision and suggest what socio-health resources are needed by the patient and their family.

## Figures and Tables

**Figure 1 ijerph-19-10366-f001:**
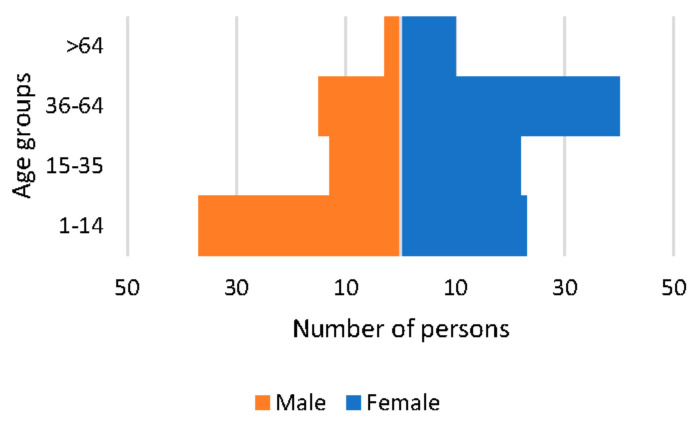
Distribution of patients by sex and age group.

**Figure 2 ijerph-19-10366-f002:**
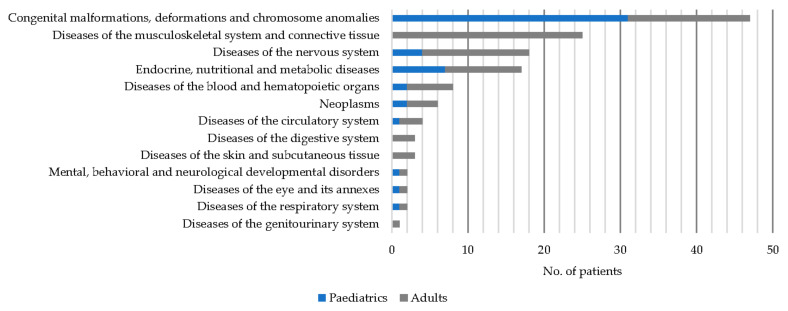
Diseases by ICD10-ES diagnostic group and patient type.

**Figure 3 ijerph-19-10366-f003:**
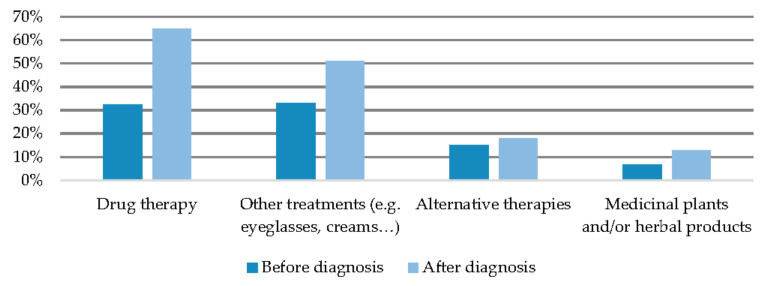
Types of treatment before and after diagnosis.

**Figure 5 ijerph-19-10366-f005:**
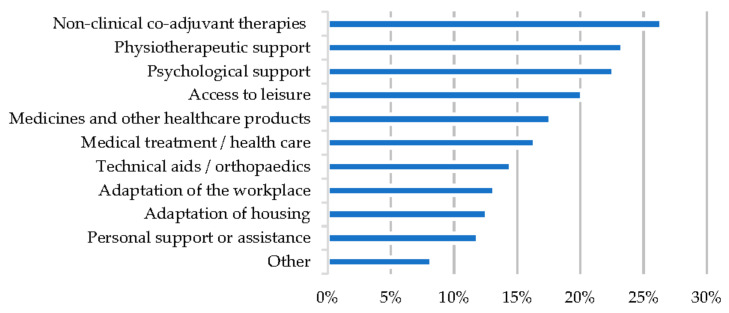
Unmet needs.

**Figure 6 ijerph-19-10366-f006:**
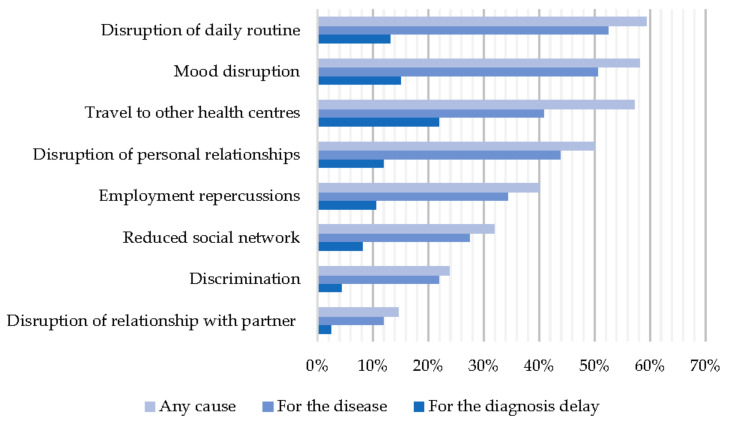
Emotional and social impact as a function of perceived cause.

**Table 1 ijerph-19-10366-t001:** Socio-demographic and diagnostic characteristics of paediatric and adult patients.

	Paediatrics			Adults			Total		
	Both Sexes	Male	Female	Both Sexes	Male	Female	Both Sexes	Male	Female
*n* (%)	60 (36.8%) **	37 (61.7%)	23 (38.3%)	103 (6.2%) **	31 (30.1%)	72 (69.9%)	163 (100%)	68 (41.7%)	95 (58.3%)
Mean age in years ± sd	8.28 ± 3.22	8.24 ± 3.38	8.35 ± 3.02	45.08 ± 18.58	40.42 ± 20.37	47.08 ± 17.53	31.53 ± 23.196	22.91 ± 21.27	37.71 ± 22.64
Confirmed diagnosis (%)	50 (83.3%)	32 (86.5%)	18 (78.3%)	88 (85.4%)	26 (83.9%)	62 (86.1%)	138 (84.7%)	58 (85.3%)	80 (84.2%)
Average age (years) at symptom onset ± sd *	1.46 ± 2.79	1.72 ± 2.99	1 ± 2.42	26.29 ± 20.69	21.30 ± 21.02	28.38 ± 20.38	18.02 ± 20.61	11.1 ± 17.58	22.83 ± 21.31
Mean age (years) at diagnosis ± sd *	2.4 ± 3.41 **	3.07 ± 3.62	1.24 ± 2.73	31.83 ± 21.17	28.73 ± 23.38	33.27 ± 20.12	21.11 ± 22.13 **	14.98 ± 20.55	25.81 ± 22.23
Median time (months) to diagnosis (IQR) *	5 (0–18)	7 (0–38.25)	0 (0–10.5)	23.5 (5.75–95)	20 (3–148.25)	23.5 (9.75–74.75)	14 (2–65)	11 (0.5–68.25)	14 (IQR 4–61.5)
Diagnostic delay (%) *	14 (29.8%)	12 (40%)	2(11.8%)	51 (62.2%)	14 (53.8%)	37 (66.1%)	65 (50.4%)	26 (46.4%)	39 (53.4%)

* Calculated from *n* = 129; ** Difference between sexes, *p*-value < 0.05; IQR: Interquartile range.

**Table 2 ijerph-19-10366-t002:** Treatments used before and after diagnosis.

		Sex (M/F)	Type of Patient (Adult/Paediatric)	Diagnostic Delay (y/n)	Hospitalisation (y/n)	Aggravation (y/n)
Before diagnosis	Drug therapy	25.9%/37.2%	39.1%/20.4% *	44.6%/20.6% *	34.7%/29.0%	42.6%/23.3% *
	Other treatments	24.1%/40.0%	35.7%/28.6%	44.4%/21.3% **	37.1%/29.0%	47.5%/21.9% **
	Alternative therapies	13.8%/15.8%	17.6%/10.2%	15.6%/13.1%	16.9%/12.9%	20.0%/11.0%
	Medicinal plants and/or herbal products	3.5%/9.2%	8.2%/4.2%	7.8%/6.7%	8.5%/4.9%	11.7%/2.8%
After diagnosis	Drug therapy	56.4%/71.1%	74.7%/47.9% **	23.4%/12.9%	75.7%/53.3% *	77.6%/55.6% *
	Other treatments	50.0%/51.9%	53.5%/46.9%	59.4%/43.5%	54.2%/47.5%	66.7%/38.4% **
	Alternative therapies	12.1%/22.1%	22.1%/10.2%	23.4%/12.9%	26.8%/8.1% **	25.0%/12.3%
	Medicinal plants and/or herbal products	12.1%/13.2%	12.9%/12.2%	12.7%/14.5%	17.1%/8.1%	22.0%/5.5% **

* *p*-value < 0.05; ** *p*-value < 0.01.

**Table 3 ijerph-19-10366-t003:** Expenditures, unmet needs, and impacts secondary to the rare disease.

		Sex (M/F)	Type of Patient (Adult/Paediatric)	Diagnostic Delay (y/n)	Hospitalisation (y/n)	Aggravation (y/n)
Expenditures *n* = 161	Drug therapy (medicines)	26.9%/50% *	48%/27.1% **	53.8%/33.9% *	51.8%/28.4% *	52.2%/31.1% *
	Therapy (physiotherapy, speech therapy, rehabilitation)	44.1%/33.7%	34%/45%	37.5%/40.6%	44.6%/29.7%	47.8%/28.9% *
	Psychological support	11.8%/17.6%	15%/15.3%	18.8%/11.1%	22.9%/6.8% **	25.4%/6.7% **
	Alternative medicine	7.4%/13.2%	10%/11.9%	12.5%/11.1%	16.9%/4.1% *	16.4%/5.6% *
	Support person	11.8%/16.5%	18%/8.5%	20.3%/11.1%	19.3%/9.5%	22.4%/8.9% *
	Specialised centre	14.7%/20.7%	15.8%/22%	20%/22.2%	22.6%/12.2%	22.1%/13.3%
Unmet needs *n* = 159	Medical treatment/health care	11.9%/19.6%	20%/10.2%	26.2%/12.9%	19.3%/13.5%	19.4%/13.3%
	Psychological support	20.9%/23.9%	20%/27.1%	24.6%/16.1%	27.7%/17.6%	31.3%/15.6% *
	Physiotherapy	23.9%/22.8%	25%/20.3%	24.6%/21%	30.1%/16.2% *	35.8%/13.3% **
	Medicines and other healthcare products	14.9%/19.6%	19%/15.3%	27.7%/12.9% *	19.3%/16.2%	22.4%/13.3%
	Orthopaedics	16.4%/13%	12%/18.6%	7.9%/8.7%	19.3%/9.5%	22.4%/7.8% **
	Personal support	10.4%/13%	14%/8.5%	10.8%/11.3%	15.7%/8.1%	17.9%/6.7% *
	Adaptation of housing	13.4%/12%	14%/10.2%	12.3%/12.9%	18.1%/6.8% *	22.4%/4.4% **
	Adaptation of workplace	9%/16.3%	15%/10.2%	16.9%/8.1%	19.3%/6.8% *	26.9%/2.2% **
	Non-clinical adjuvant therapies	25.4%/27.2%	24%/30.5%	26.2%/25.8%	36.1%/16.2% **	44.8%/12.2% **
	Leisure and free time	19.4%/20.7%	20%/20.3%	20%/21%	27.7%/12.2% *	32.8%/10% **
	Other	7.5%/8.8%	7%/10.3%	4.6%/9.8%	9.8%/6.8%	11.9%/5.6%
Impacts *n* = 155	Travel to other health centres	55.9%/55.2%	58%/55.9%	64.1%/60.3%	71.4%/41.1% **	73.5%/43.8% **
	Employment repercussions	32.4%/45.7%	44.6%/32.2%	43.1%/41.3%	57.1%/21.6% **	58.8%/25.6% **
	Disruption of personal relationships	41.2%/56.5%	55.4%/40.7%	55.4%/50.8%	69%/28.4% **	76.5%/28.9% **
	Disruption of relationship with partner	16.4%/13.2%	17%/10.3%	24.6%/11.5%	23.2%/5.4% **	20.9%/10%
	Mood disruption	48.5%/65.2% *	65.3%/45.8% *	56.9%/63.5%	72.6%/40.5% **	79.4%/41.1% **
	Disruption of daily routine	58.8%/59.8%	62.4%/54.2%	61.5%/65.1%	76.2%/40.5% **	80.9%/43.3% **
	Reduced social network	25%/37%	36.6%/23.7%	36.9%/30.2%	44%/18.9% **	50%/17.8% **
	Discrimination	22.1%/25%	25.7%/20.3%	27.7%/25.4%	32.1%/14.9% *	33.8%/16.7% *

* *p*-value < 0.05; ** *p*-value < 0.01.

## Data Availability

The data presented in this study are available on request from the corresponding author. The data are not publicly available due to ethical restrictions.

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
