# Peer review of "Rare Diseases: Needs and Impact for Patients and Families: A Cross-Sectional Study in the Valencian Region, Spain"

_ijerph, 2022, doi:10.3390/ijerph191610366_

Round 1

Reviewer 1 Report

In this manuscript, the authors have described a cross-sectional study of rare diseases in the region of Valencia, Spain. It seems that what makes rare diseases difficult to diagnose and to some degree begin their treatment is the lack of definition of what constitutes a rare disease. I would have liked if the authors made a few points on their interpretation of what constitutes a rare disease. The manuscript is well-written, and it is easy to follow. The graphs are supported by the text in the manuscript.

Minor points:

Why did the authors combine pediatric and adult rare diseases in this manuscript? Would it be convenience or because they expected different pathologies/classifications if broken down into these two categories?

Would it be possible to make the graphs black instead of gray? It is a bit difficult to see.

Why was there a lack of elderly people in your sample pool? Were they under-represented or rare diseases do not reach this population?

 Would it be possible to group rare diseases by pathologies and this way, the analysis would be disease-specific?

Are the authors planning to carry out a follow-up study to determine if their identified barriers are improved because of this knowledge?

Reviewer 2 Report

The topic of the manuscript is very important and still there is little research on it. The study is well designed, and the methodology is well suited.  The conclusions are drawn based on the results. The article can be accepted in its present form. I have some minor comments.

1. For clarity please define rare diseases. If there was still no diagnosis how were the rare diseases defined?

2. Please include one sentence about what kind of services are paid for by the government. Are psychological therapies private or included in public healthcare?

3. Conclusions-training health professionals on socio-health aspects: is there a position of social assistant in Spanish healthcare who can work with patients on these issues?

Reviewer 3 Report

The article is on the whole interesting, it deals with a very important public health issue and is structured in an effective and coherent way. I consider this article worthy of publication with some minor revisions.

Here, my comments:

- Line 74, it would be useful for the reader to know which specific areas have been explored.

- Line 79-83: the Authors state that this study is part of a bigger study, please give more information on this, for example funding, aim of the bigger project, future steps of the bigger project.

- In the Materials and Methods section there are no mention of the submission of the project to any ethical committee/board, please specify if you obtained the approval of the competent ethical committee and, otherwise, the reasons why you didn’t.

- In paragraph 3.1 please specify what you mean with “alternative therapies”. It is not clear to me the reason why you included “creams” in “other therapies”, since pharmacological treatments with topical medications are of course “drug therapy” (think, for example, to corticosteroid ointments); moreover, eyeglasses are not “treatments”, they are medical devices that may mitigate symptoms and help patients in their daily life. I suggest the Authors to better clarify these different types of “treatments” cited in par 3.1.

- Line 443 you refer to the issue of equity in healthcare. This point is of great importance in any dissertation about determinants of health. I think that a deeper reference to the issue of equity and equality in healthcare should be provided in the introduction in order to introduce the reader to this sub-topic.

- I believe that the final part of the discussion could benefit from some suggestions that the authors would like to propose to improve the perception of these fragile patients. It could be useful to provide a bulleted list of strategic indications both for policy makers and public health administrators and for researchers (future perspectives, gaps to be filled, etc.).
